# RangePerception: Taming LiDAR Range View for Efficient and Accurate 3D Object Detection

**Yeqi Bai**[1]    **Ben Fei**[1,2]    **Youquan Liu**[1]    **Tao Ma**[1,3]
**Yuenan Hou**[1]    **Botian Shi**[1,*]    **Yikang Li**[1]
[1]Shanghai AI Laboratory    [2]Fudan University
[3]Multimedia Laboratory, The Chinese University of Hong Kong
{baiyeqi,feiben,liuyouquan,matao}@pjlab.org.cn
{houyuenan,shibotian,liyikang}@pjlab.org.cn

## Abstract

LiDAR-based 3D detection methods currently use bird's-eye view (BEV) or range view (RV) as their primary basis. The former relies on voxelization and 3D convolutions, resulting in inefficient training and inference processes. Conversely, RV-based methods demonstrate higher efficiency due to their compactness and compatibility with 2D convolutions, but their performance still trails behind that of BEV-based methods. To eliminate this performance gap while preserving the efficiency of RV-based methods, this study presents an efficient and accurate RV-based 3D object detection framework termed *RangePerception*. Through meticulous analysis, this study identifies two critical challenges impeding the performance of existing RV-based methods: 1) there exists a natural domain gap between the 3D world coordinate used in output and 2D range image coordinate used in input, generating difficulty in information extraction from range images; 2) native range images suffer from vision corruption issue, affecting the detection accuracy of the objects located on the margins of the range images. To address the key challenges above, we propose two novel algorithms named Range Aware Kernel (RAK) and Vision Restoration Module (VRM), which facilitate information flow from range image representation and world-coordinate 3D detection results. With the help of RAK and VRM, our *RangePerception* achieves 3.25/4.18 higher averaged L1/L2 AP compared to previous state-of-the-art RV-based method RangeDet, on Waymo Open Dataset. For the *first time* as an RV-based 3D detection method, *RangePerception* achieves slightly superior averaged AP compared with the well-known BEV-based method CenterPoint and the inference speed of *RangePerception* is 1.3 times as fast as CenterPoint.

## 1   Introduction

In recent years, LiDAR-based 3D perception [1, 2, 3, 4, 5] has made tremendous advances in the field of autonomous driving. One primary task in this area is 3D detection [1, 2, 6, 7, 8, 9, 10], which involves predicting the 3D locations of objects of interest, as well as their geometric and motional properties, including categories, 3D sizes, headings, and velocities. Despite sharing some similarities, there is a fundamental difference between LiDAR-based 3D detection and image-based 2D detection [11, 12, 13, 14, 15, 16]: RGB images are inherently well-structured matrices, whereas LiDAR signals are sets of sparse and unordered points in 3D space. Considering modern computer vision techniques, such as Convolutional Neural Networks (CNNs) [17, 18] and Vision Transformers

---

*Corresponding author

37th Conference on Neural Information Processing Systems (NeurIPS 2023).

(ViTs) [19, 20] require well-structured matrices as inputs, an essential process in LiDAR-based 3D detection is to effectively organize the unstructured LiDAR points.

To effectively organize LiDAR points and facilitate the use of established computer vision techniques, two major representations have been adopted: bird's-eye view (BEV) and range view (RV). Shown in Fig. 1(a-d) is a frame of top LiDAR signal from Waymo Open Dataset (WOD) [21], represented in RV and BEV accordingly. BEV-based 3D detectors [1, 2, 6] convert sparse LiDAR points into 3D voxel grids, extract features with a 3D convolution backbone, and perform classification and regression after mapping the extracted 3D features to the BEV. With the aid of well-organized voxel representation and compatibility with well-developed detection paradigms, BEV-based methods exhibit the highest detection accuracy among contemporary LiDAR-based 3D object detectors. However, it is worth pointing out that a LiDAR sensor in autonomous driving scenarios [22, 23, 24] uniformly samples from a spherical coordinate system, and the cartesian-coordinated voxel representation has certain incompatibility with LiDAR sensors, as evidenced by two factors. First, as the distance from LiDAR grows longer, the distribution of voxels becomes a lot sparser, despite that LiDAR beams are uniformly distributed in the spherical coordinate. Second, when the horizontal length of farthest beam extends to the time of $l$, the space of the voxel representation has to increase to the time of $l^2$ to contain complete information. As a result, BEV-based methods have certain drawbacks in their applications, such as the requirement for complex and time-consuming sparse 3D convolutions as their backbones, limiting their efficiency. Moreover, if users aim to augment horizontal perception range, the computational complexity of BEV-based detectors must increase quadratically, which poses a challenge for real-time implementation.

The range-view (RV) representation, on the other hand, is naturally generated from the scanning mechanism of the LiDAR sensor [25, 26, 27, 28]. Each pixel in the range image corresponds to an optical response of the LiDAR beam, making the range view the most compact and informative way to represent LiDAR signals. The compactness of range images also enables RV-based 3D detectors [29, 30, 9, 10] to enjoy more compact feature spaces and higher inference speeds, compared to the BEV-based 3D detectors. However, in the aspect of detection accuracy, pioneering RV-based 3D detectors significantly lag behind the top-performing BEV-based detectors, with a performance gap of more than 10 average L1 AP on WOD validation set. More recently, RangeDet [9] proposed several modules with stronger representation power, narrowing the performance gap to 2.77 average L1 3D AP on WOD, compared with state-of-the-art BEV-based method CenterPoint [6]. On top of the arts above, FCOS-LiDAR [10] develops a multi-frame RV-based detection pipeline on nuScenes dataset [31, 32]. Despite being 90% faster than CenterPoint, FCOS-LiDAR's overall validation AP is 3.32 lower, evaluated with multi-frame setting on nuScenes dataset.

To better exploit the potential of the range-view representation, a detailed analysis of existing RV-based detection methods is conducted, which reveals two critical unsolved challenges.

**Spatial Misalignment**. Existing RV-based detectors treat range images the same way as RGB images, by directly feeding them into 2D convolution backbones. This workflow neglects the nature that range images contain rich depth information, and even two range pixels are adjacent in range coordinate, their actual distance in 3D space could be more than 30 meters. As visualized in Fig. 1(e), foreground pixels on the margins of vehicles and pedestrians are often far from their neighboring background pixels in 3D space. Directly processing such 3D-space-uncorrelated pixels with 2D convolution kernels can only produce noisy features, hindering geometric information extraction from the margins of foreground objects. This phenomenon will be termed as Spatial Misalignment in further discussion of this paper.

**Vision Corruption**. When objects of interest are located on the margins of range images, as shown in Fig. 1(c,f), their corresponding foreground range pixels are separately distributed around the left and right borders of the range image. Since CNNs have limited receptive fields, features around the left border cannot be shared with features around the right border and vice versa, when 2D convolution backbones are used as feature extractors. This phenomenon, called Vision Corruption, can significantly impact the detection accuracy of objects on the margins of range images. Previous RV-based detection methods have overlooked this issue and directly processed range images with 2D convolution backbones without compensating for the corrupted areas.

In this paper, we demonstrate an efficient and accurate RV-based 3D detection framework, termed *RangePerception*. To overcome the key challenges above, two novel algorithms named *Range Aware Kernel (RAK)* and *Vision Restoration Module (VRM)* are proposed and integrated into *RangePer-*

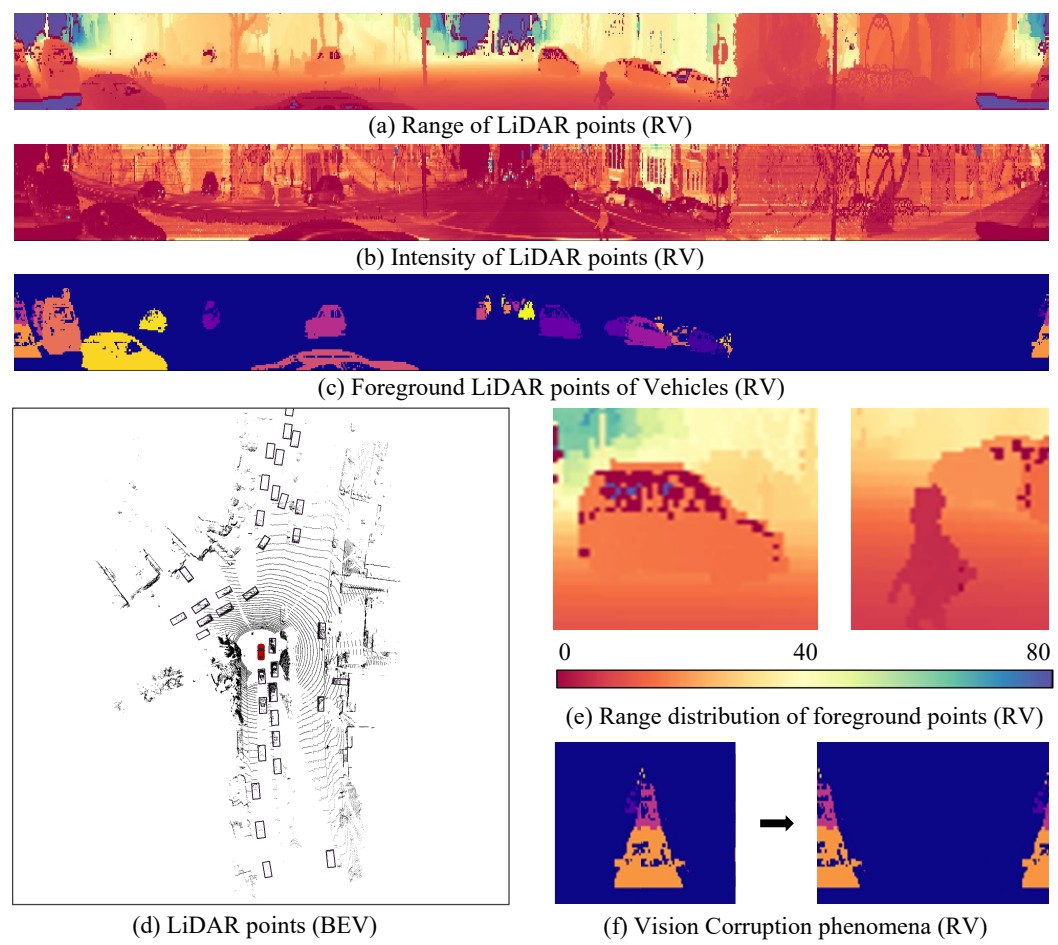

(a) Range of LiDAR points (RV)

(b) Intensity of LiDAR points (RV)

(c) Foreground LiDAR points of Vehicles (RV)

0   40   80

(e) Range distribution of foreground points (RV)

(d) LiDAR points (BEV)

(f) Vision Corruption phenomena (RV)

Figure 1: (a-d) A sample frame of top LiDAR signal, represented in RV and BEV respectively. (e) Spatial Misalignment phenomena. (f) Vision Corruption phenomena.

*ception* framework, both facilitating information flow from range image representation and world-coordinate 3D detection results. With the help of RAK and VRM, our *RangePerception* presents 73.62 & 80.24 & 70.33 L1 3D AP for vehicle & pedestrian & cyclist, on WOD, achieving state-of-the-art performance as a range-view-based 3D detection method. The contributions of this paper are presented as follows.

**RangePerception Framework.** A novel high-performing 3D detection framework, named RangePerception, is introduced in this paper. RangePerception is the first RV-based 3D detector to achieve 74.73/69.17 average L1/L2 AP on WOD, outperforming the previous state-of-the-art RV-based detector RangeDet, which has average L1/L2 APs of 71.48/64.99, presenting an improvement of 3.25/4.18. RangePerception also demonstrates slightly superior performance compared to widely-used BEV-based method CenterPoint [6], which has average L1/L2 APs of 74.25/68.04. Notably, RangePerception's inference speed is 1.3 times as fast as CenterPoint, justifying better fitness for real-time deployment on autonomous vehicles.[2]

**Range Aware Kernel.** As part of RangePerception's feature extractor, Range Aware Kernel (RAK) is a trailblazing algorithm tailored to RV-based networks. RAK disentangles the range image space into multiple subspaces, and overcomes the Spatial Misalignment issue by enabling independent feature extraction from each subspace. Experimental results show that RAK lifts the average L1/L2 AP by 5.75/5.99, while incurring negligible computational cost.

---

[2]Project website is available at `https://rangeperception.github.io`, to enhance the accessibility and comprehension of this study.

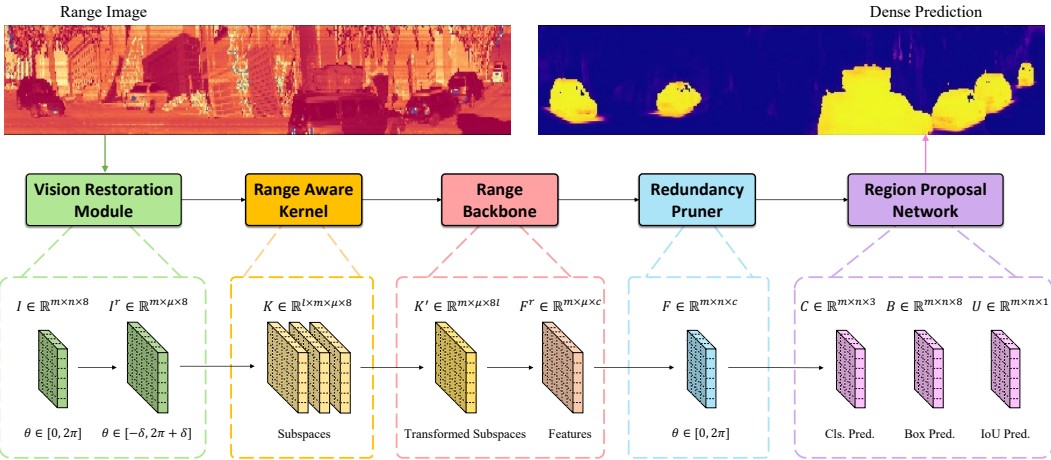

Figure 2: The RangePerception framework takes a range image $I$ as input and generates dense predictions. To improve representation learning, the framework sequentially integrates the VRM and RAK modules before the Range Backbone. Subsequently, a specially devised Redundancy Pruner is used to remove redundancies in the deep features, thereby mitigating the computational cost in the subsequent Region Proposal Network and post-processing layers.

**Vision Restoration Module.** To resolve the Vision Corruption issue, Vision Restoration Module (VRM) is brought to light in this study. VRM extends the receptive field of the backbone network by restoring previously corrupted areas. VRM is particularly helpful to the detection of vehicles, as will be illustrated in the experiment section.

## 2 Preliminary of Range-view Representation

This section provides a brief overview of the range view representation of LiDAR data. Specifically, LiDAR data can be represented as an $m \times n$ matrix, known as a range image, where $m$ represents the number of beams emitted and $n$ represents the number of measurements taken during one scan cycle. Each column of the range image corresponds to a shared azimuth, while each row corresponds to a shared inclination, indicating the relative vertical and horizontal angles of a returned point with respect to the LiDAR's original point. Each pixel in the range image contains at least three geometric values, namely range $r$, azimuth $\theta$, and inclination $\phi$, which define a spherical coordinate system. The widely-used point cloud data with Cartesian coordinates is derived from the spherical coordinate system: $x = r\cos(\phi)\cos(\theta), y = r\cos(\phi)\sin(\theta), z = r\sin(\phi)$, where $x$, $y$, $z$ denote the Cartesian coordinates of the points. Considering modern LiDAR sensors [33, 34, 35] often measure magnitude of the returned laser pulse named intensity $\eta$ and elongation $\rho$, range view of LiDAR signal can be engineered as $I \in \mathbb{R}^{m \times n \times 8}$, with each pixel being $I_{j,k} = \{r, x, y, z, \theta, \phi, \eta, \rho\}$. This study additionally defines range matrix as $R \in \mathbb{R}^{m \times n}$, with each pixel being $R_{j,k} = r$, for the sake of further discussion. To provide a better illustration, Fig. 1(a) presents a sample frame of range matrix $R$ and Fig. 1(b) shows intensity values in the corresponding range image $I$.

## 3 Methodology

The RangePerception framework, depicted in Fig. 2, takes range image $I$ as input and produces dense predictions. To enhance representation learning, VRM and RAK are sequentially incorporated before the Range Backbone. Afterwards, a carefully designed Redundancy Pruner is employed to eliminate redundancies in deep features, which minimizes computational costs in subsequent Region Proposal Network (RPN) and post-processing layers. This section begins with comprehensive explanations of RAK and VRM, followed by an in-depth elaboration of RangePerception architecture.

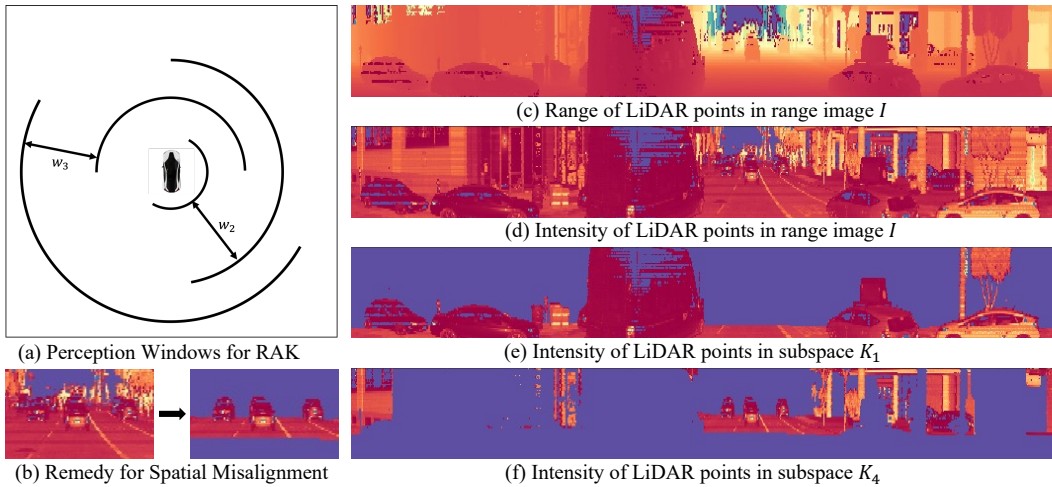

(a) Perception Windows for RAK

(c) Range of LiDAR points in range image $I$

(d) Intensity of LiDAR points in range image $I$

(e) Intensity of LiDAR points in subspace $K_1$

(b) Remedy for Spatial Misalignment

(f) Intensity of LiDAR points in subspace $K_4$

Figure 3: Range Aware Kernel disentangles the range image space into multiple subspaces, and overcomes the Spatial Misalignment issue by enabling independent feature extraction from each subspace.

## 3.1 Range Aware Kernel

As a key component of RangePerception's feature extractor, Range Aware Kernel is an innovative algorithm specifically designed for RV-based networks. RAK disentangles the range image space into multiple subspaces, and overcomes the Spatial Misalignment issue by enabling independent feature extraction from each subspace. Shown in Fig. 3(a), Range Aware Kernel comes with a set of $l$ predefined Perception Windows, formulated as $W = \{w_1, w_2, ..., w_{l-1}, w_l\}$, where each Perception Window is a range-conditioned interval $w_i = [r_{i1}, r_{i2}]$.

Given a frame of range image $I \in \mathbb{R}^{m \times n \times 8}$, RAK first calculates binary mask $M \in \mathbb{Z}^{l \times m \times n \times 8}$ according to Perception Windows $W = \{w_1, w_2, ..., w_{l-1}, w_l\}$, where each $M_i \in \mathbb{Z}^{m \times n \times 8}$ is a pixel-wise mask for range image $I$, indicating whether each range value $R_{j,k} = I_{j,k,1}$ stays in current Perception Window $w_i$. Subsequently, RAK defines a tensor $K \in \mathbb{R}^{l \times m \times n \times 8}$ representing $l$ subspaces, and derives each subspace $K_i = M_i \odot I$. Detailed computing logic of RAK is illustrated in Algorithm 1, note that though inference process of RAK seems to incur $O(lmn)$ time complexity, GPU implementation of RAK can readily achieve $O(1)$ with proper parallelism.

As elaborated above, RAK divides range image $I$ into multiple subspaces $K$, where each subspace $K_i$ contains LiDAR points that belong to Perception Window $w_i$. To provide a clearer visualization, Fig. 3(c,d) displays range and intensity values of a frame of input range image $I$. By further processing range image $I$ with RAK, tensor $K$ is computed, from which intensity values of subspaces $K_1$ and $K_4$ are presented in Fig. 3(e,f). RAK effectively separates foreground vehicle points from their background counterparts, thus minimizing Spatial Misalignment and facilitating feature extraction from range view. Fig. 3(b) provides further evidence of the efficacy of RAK, by clearly disentangling previously indistinguishable vehicles from noisy background points.

In the architecture of RangePerception, Range Aware Kernel is positioned directly before backbone network. Subspaces $K \in \mathbb{R}^{l \times m \times n \times 8}$, generated from Range Aware Kernel, is subsequently fed into the backbone network for non-linear feature extraction. Experimental results demonstrate that RAK increases the average L1/L2 AP by 5.75/5.99, while incurring negligible computational cost.

## 3.2 Vision Restoration Module

As described in Sec. 2, each column in range image $I$ corresponds to a shared azimuth $\theta \in [0, 2\pi]$, indicating the spinning angle of LiDAR. Specifically, $\theta = 0$ at left margin of range image and $\theta = 2\pi$ at right margin of range image. Due to the periodicity of LiDAR's scanning cycle, azimuth values $0$ and $2\pi$ correspond to beginning and end of each scanning cycle, both pointing in the opposite

**Algorithm 1** Range Aware Kernel

1: **input** range image $I \in \mathbb{R}^{m \times n \times 8}$
2: **init** range matrix $R \in \mathbb{R}^{m \times n}$
3: **init** $i, j, k \in \mathbb{Z}$
4: **for** $j \in [1, m], k \in [1, n]$ **do**
5:     $R_{j,k} \leftarrow I_{j,k,1}$
6: **init** tensor $K \in \mathbb{R}^{l \times m \times n \times 8}$
7: **init** binary mask $M \in \mathbb{Z}^{l \times m \times n \times 8}$
8: **for** $w_i \in W$ **do**
9:     **for** $R_{j,k}$ **in** $R$ **do**
10:         **if** $R_{j,k} \in w_i$ **then**
11:             $M_{i,j,k} \leftarrow \mathbf{1}$
12:         **else if** $R_{j,k} \notin w_i$ **then**
13:             $M_{i,j,k} \leftarrow \mathbf{0}$
14:     $K_i \leftarrow M_i \odot I$
15: **output** tensor $K$

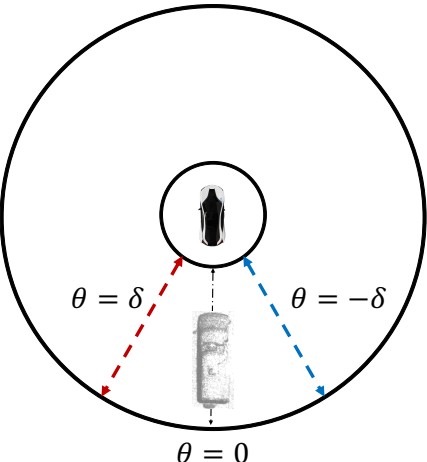

Figure 4: Spherical Coordinate of VRM.

direction of the ego vehicle. As illustrated in Fig. 4, objects located behind ego vehicle are often separated by ray with $\theta = 0$, resulting in Vision Corruption phenomena elaborated in Fig. 1(c,f).

To resolve Vision Corruption issue, Vision Restoration Module is brought to light in this study. By predefining a restoration angle $\delta$, VRM builds an extended spherical space with azimuth $\theta \in [-\delta, 2\pi + \delta]$. In this way, visual features originally corrupted by LiDAR's sampling process are restored on both sides of range image $I$, significantly easing the feature extraction from the margins of $I$. This restoration process is clearly visualized in Fig. 5(a,b), where VRM-extended range images can be termed as $I^r$. In the architecture of RangePerception, restored range images $I^r$ are subsequently processed by RAK and range backbone for deeper pattern recognition. Note that in Sec. 3.1, input range image of RAK is still denoted as $I$ instead of $I^r$, for the sake of simplicity.

It is straightforward to observe from Fig. 5(a,b) that VRM introduces redundancies to range view: region with azimuth $\theta \in [-\delta, \delta]$, appears twice in restored range image $I^r$. Though this duplication helps information extraction from the margins of $I$, redundancies persist in feature space $F^r$ learnt by RAK and range backbone, leading to unnecessary computational costs for subsequent region proposal network and post-processing layers. To address this issue and improve efficiency, Redundancy Pruner (RP) is designed and equipped with RangePerception framework. Operating on feature space $F^r$, RP performs inverse function of VRM, by pruning $F^r$ from azimuth interval $[-\delta, 2\pi + \delta]$ back to $[0, 2\pi]$.

To better explain the process above, a pseudo VRM-extended image $I^r$ is rendered in Fig. 5(c). Vanilla pixels in $I$ are filled with zeros, while pixels generated by VRM are filled with ones. A pseudo feature space $F^r$ is subsequently computed, via inputting pseudo image $I^r$ to RAK and backbone. Finally, RP drops redundancies in pseudo $F^r$, resulting in pruned feature space $F$. As observed from lower part of Fig. 5(c), visual information that belongs to VRM-restored spaces readily flows to vanilla space, thanks to the layered convolution kernels. This justifies that Redundancy Pruner solely decreases computational costs, without causing any loss in visual features.

### 3.3 RangePerception Framework

For the sake of efficiency, RangePerception is designed as an anchor-free single-stage detector, as presented in Fig. 2. Given a frame of range image $I$, RangePerception framework first compensates the corrupted regions with VRM, resulting in restored image $I^r \in \mathbb{R}^{m \times \mu \times 8}$, where $\mu = \frac{m(\delta + \pi)}{\pi}$. Subsequently, RAK converts $I^r$ into subspaces $K \in \mathbb{R}^{l \times m \times \mu \times 8}$, disentangling misaligned pixels. Range Backbone, derived from DLA-34 [36], is adopted to extract non-linear features $F^r \in \mathbb{R}^{m \times \mu \times c}$ from subspaces. Further, RP eliminates redundant features, generating pruned features $F \in \mathbb{R}^{m \times n \times c}$. RPN learns dense predictions $\{C, B, U\}$ on top of deep features $F$, representing class predictions, box predictions, and IoU predictions accordingly. Finally, Weight NMS is employed to aggregate dense predictions, where IoU predictions $U$ are treated as per-box confidence scores.

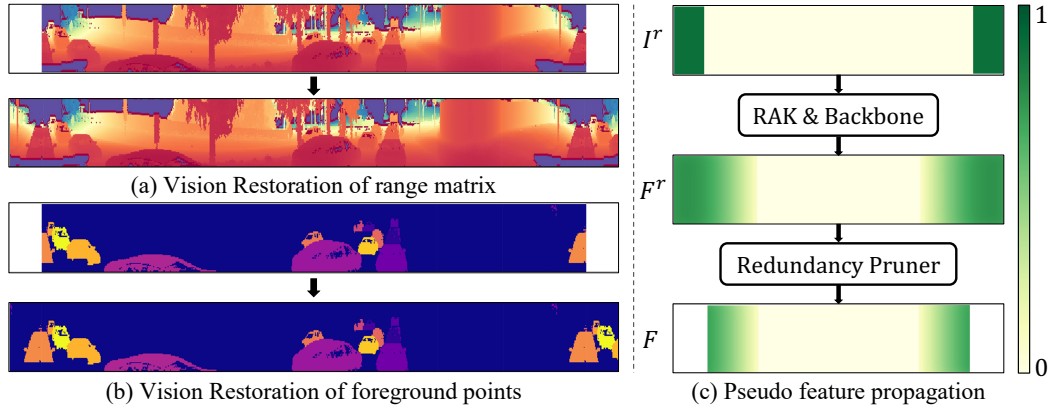

(a) Vision Restoration of range matrix

(b) Vision Restoration of foreground points

(c) Pseudo feature propagation

Figure 5: Vision Restoration Module. By predefining a restoration angle $\delta$, VRM builds an extended spherical space with azimuth $\theta \in [-\delta, 2\pi + \delta]$. As a result, Vision Corruption issues are resolved on both sides of range image $I$, significantly easing the feature extraction from the margins of $I$.

## 4 Related Work

**BEV-based 3D Detection.** The majority of the highest-performing LiDAR-based 3D detectors are categorized as BEV-based detection, where the initial step involves the conversion of the point cloud into BEV (Bird's Eye View) images. VoxelNet [37] is the pioneering end-to-end BEV-based detector that utilizes PointNet [38] for inner-voxel representation and 3D convolutions for high-level feature generation, where high-level features are further processed by region proposal network (RPN). To reduce the computational burden of 3D convolutions, SECOND [1] introduces the usage of sparse convolutions. Another common approach is to eliminate voxelization along the elevation axis and convert the point cloud into pillars instead of voxels, as proposed by [2]. By leveraging either voxel-based or pillar-based BEV representation, CenterPoint [6] achieves state-of-the-art performance levels in a center-based anchor-free manner. However, complex and time-consuming sparse 3D convolutions in these methods hinder their practical applications, and lightweight 3D detectors urgently need to be developed.

**Multi-view-based 3D Detection.** Most top-performing multi-view-based (MV-based) 3D detectors [7, 8] adopt a two-stage approach, where the first stage typically employs a BEV-based detector, and point-view features are subsequently utilized in the second stage for proposal refinement. PV-RCNN [7] combines 3D voxel CNN and PointNet-based [38] set abstraction to learn discriminative features from point clouds. Part-$A^2$ [8] introduces part-awareness multi-view aggregation, achieving outstanding performance by sequentially predicting coarse 3D proposals and accurate intra-object part locations. However, despite the excellent detection performance achieved by MV-based methods, their inference speed remains impractical for real-time deployment on vehicles due to the computational complexity of their structure.

**Range-view-based 3D Detection.** In light of the compactness of the RV representation, certain approaches endeavor to perform detection based on RV. VeloFCN [39] was the first work to perform 3D object detection using RV, which involves transforming the point cloud to a range image and then applying 2D convolution to detect 3D objects. Subsequent research endeavors [29, 30, 9, 10] are put forth to improve the efficacy of RV-based detectors. For instance, LaserNet [29] models the distribution of 3D box corners to capture their uncertainty, resulting in more accurate detections. RCD [30] introduces the range-conditioned dilation mechanism to dynamically adjust the dilation rate based on the measured range, thereby alleviating the scale-sensitivity issue of RV-based detectors. Moreover, RangeDet [9] proposes the Range Conditioned Pyramid to mitigate the scale-variation issue and utilizes the Meta-Kernel convolution to better exploit the 3D geometric information of the points. Despite outperforming all previous RV-based methods, RangeDet lags behind the widely-used BEV-based method CenterPoint by 2.77 average AP, evaluated on WOD. Recently, FCOS-LiDAR [10] proposes a novel range view projection mechanism, and demonstrates the benefits of fusing multi-frame point clouds for a range-view-based detector. Although FCOS-LiDAR is 90% faster than CenterPoint, its overall validation AP is 3.32 lower, evaluated with multi-frame setting on nuScenes dataset.

Table 1: Detection performance measured by 3D AP/APH on WOD validation set, along with inference speed measured by FPS. All experiments are conducted under single-frame setting.

| Method | View | Stage | Vehicle | | Pedestrian | | Cyclist | | Average | | FPS |
|---|---|---|---|---|---|---|---|---|---|---|---|
| | | | L1 | L2 | L1 | L2 | L1 | L2 | L1 | L2 | |
| Second | B | one | 72.25/71.67 | 63.84/63.32 | 68.69/58.12 | 60.73/51.26 | 60.61/59.27 | 58.35/57.03 | 67.18/63.02 | 60.97/57.20 | 29.51 |
| PointPillar | B | one | 71.76/71.25 | 63.50/62.93 | 69.65/47.41 | 61.68/41.27 | 59.03/52.95 | 56.81/51.38 | 66.81/57.20 | 60.66/51.65 | 41.64 |
| CenterPoint | B | one | 74.58/74.04 | 66.42/65.93 | 76.17/69.98 | 68.33/62.63 | **72.00/70.80** | **69.39/68.23** | 74.25/71.61 | 68.05/65.60 | 34.73 |
| PV-RCNN | B+P | two | 76.70/76.11 | 68.44/67.95 | 73.95/63.20 | 65.68/55.86 | 67.95/66.25 | 65.54/63.79 | 73.47/68.52 | 66.55/62.53 | 3.17 |
| Part-$A^2$-anchor | B+P | two | **77.03/76.49** | **68.46/67.98** | 75.26/66.89 | 66.19/58.63 | 68.61/67.37 | 66.13/64.93 | 73.63/70.25 | 66.93/63.85 | 8.25 |
| RangeDet | R | one | 72.85/72.33 | 64.03/63.57 | 75.94/71.94 | 67.60/63.89 | 65.67/64.39 | 63.33/62.08 | 71.48/69.55 | 64.99/63.18 | 34.88 |
| RangePerception | R | one | 73.62/73.11 | 66.47/66.00 | **80.24/76.12** | **72.29/68.54** | 70.33/68.93 | 68.75/67.43 | **74.73/72.72** | **69.17/67.32** | **45.85** |

# 5 Experiments

**Dataset.** The experiments in this study utilize the WOD dataset, which is the only dataset that provides native range images. WOD consists of 798 training sequences and 202 validation sequences, with each sequence containing approximately 200 frames. The 64-beam top LiDAR signals are utilized to train and evaluate the RangePerception and baseline models. Scan per cycle of WOD's top LiDAR is 2650, resulting in range image represented by $I \in \mathbb{R}^{64 \times 2650 \times 8}$. The metrics of L1/L2 3D AP/APH are calculated and reported following the official evaluation protocol of WOD.

**Data Augmentation.** Data augmentation techniques are employed during training to improve the model's generalization capabilities. Range images and point clouds are randomly flipped along both the x and y axes and rotated within the range of $[-\pi/4, \pi/4]$. Additionally, a random global scaling factor between $[0.95, 1.05]$ is applied. The ground-truth copy-paste data augmentation [1] approach is also utilized.

**Implementation Details.** The RangePerception framework is implemented on top of OpenPCDet codebase [40]. Since OpenPCDet only supports voxel-based and point-based models, range-view data pipeline and detection models are built from scratch. For Vision Restoration Module, restoration angle $\delta$ is predefined as $0.086\pi$, generating $I^r \in \mathbb{R}^{64 \times 2880 \times 8}$. For Range Aware Kernel, six Perception Windows are adopted: $W = \{[0, 15], [10, 20], [15, 30], [20, 40], [30, 60], [45, \infty)\}$, resulting in Subspaces $K \in \mathbb{R}^{6 \times 64 \times 2880 \times 8}$ and transformed Subspaces $K' \in \mathbb{R}^{64 \times 2880 \times 48}$. DLA-34 network is adopted as Range Backbone, by updating the input convolution kernel's fan in to $48$ channels. Models are learned using Adam optimizer with an initial learning rate of $3e-3$, scheduled with the one-cycle learning rate policy. The decay weight is set to $0.01$, and momentum range is $[0.95, 0.85]$. All models are trained with 30 epochs on WOD training set, where batch size is 32 and frame sampling rate is $100\%$. Inference speed is examined with one NVIDIA A100 GPU with batch size set to 1.

**Baseline Methods.** As shown in Table 1, state-of-the-art BEV-based [1, 2, 6], MV-based [7, 8], and RV-based [9] detectors are selected as baseline methods. All BEV-based and MV-based baselines are trained and evaluated with OpenPCDet's official PyTorch implementation. Since official open-sourced version of RangeDet is coded with MxNet [41], we reimplement RangeDet with PyTorch and integrate RangeDet-PyTorch into OpenPCDet framework. We train RangeDet-PyTorch according to settings presented in their paper and measure its inference speed under OpenPCDet framework. The detection AP/APH of RangeDet is listed according to experimental results in their paper.

**Main Results.** Detection performance measured by 3D AP/APH is reported in Table 1, where RangePerception is compared against state-of-the-art BEV-based (B), RV-based (R), and MV-based (B+P) methods. Inference speed measured by frame per second (FPS) is also presented. It is evident that RangePerception's average AP/APH surpass all baselines, which highlights the strong detection functionality of RangePerception framework. Furthermore, RangePerception achieves state-of-the-art performance in pedestrian AP/APH. We attribute this to the fact that range images better preserve visual features of small objects, while voxelization introduces quantization errors to originally sparse foreground points. The results also demonstrate that RangePerception has the fastest inference speed among all methods. Specifically, RangePerception is 1.32 times as fast as CenterPoint, which is already a highly efficient BEV-based detector. Additionally, RangePerception is the first RV-based detector to achieve higher average AP/APH compared to CenterPoint, outperforming the previous state-of-the-art RV-based method by a large margin.

**Ablation Study.** Presented in Table 2, an ablation study is conducted to assess the effectiveness of our proposed designs, specifically RAK and VRM. Firstly, the study investigates the impact of RAK

Table 2: Ablation study of RangePerception, measured by 3D AP/APH on WOD validation set.

| | Setting | Vehicle | | Pedestrian | | Cyclist | | Average | |
|---|---|---|---|---|---|---|---|---|---|
| | | L1 | L2 | L1 | L2 | L1 | L2 | L1 | L2 |
| A1 | without RAK | 70.46/70.01 | 63.45/62.89 | 72.86/68.88 | 64.56/60.72 | 63.63/62.43 | 61.44/60.25 | 68.98/67.11 | 63.18/61.29 |
| A2 | 2D Convolution | 70.48/70.03 | 63.46/62.90 | 72.86/68.87 | 64.56/60.71 | 63.64/62.43 | 61.45/60.26 | 68.99/67.11 | 63.16/61.29 |
| A3 | Meta-Kernel | 72.95/72.43 | 65.98/65.37 | 75.95/71.96 | 67.63/63.90 | 66.76/65.46 | 64.57/63.38 | 71.88/69.95 | 66.06/64.21 |
| A4 | 4 Perception Windows | 73.13/72.62 | 65.97/65.51 | 80.15/76.03 | 72.17/68.42 | 70.15/68.85 | 68.42/67.10 | 74.48/72.50 | 68.85/67.01 |
| A5 | 8 Perception Windows | 73.59/73.08 | 66.45/65.98 | **80.25/76.13** | **72.30/68.56** | 70.31/68.92 | 68.74/67.42 | 74.72/72.71 | 69.16/67.32 |
| A6 | without VRM & RP | 72.50/71.97 | 66.40/65.91 | 80.20/76.08 | 72.27/68.53 | 70.31/68.92 | 68.74/67.42 | 74.34/72.32 | 69.13/67.29 |
| | RangePerception | **73.62/73.11** | **66.47/66.00** | 80.24/76.12 | 72.29/68.54 | **70.33/68.93** | **68.75/67.43** | **74.73/72.72** | **69.17/67.32** |

by removing it from our framework (A1) and replacing it with a $1\times1$ Convolution layer of equal dimension (A2). The results show a significant decrease of more than 5 average L1 AP for both cases, highlighting the crucial role of RAK in achieving strong detection performance. Secondly, we compare the performance of RAK and Meta-Kernel by replacing RAK with Meta-Kernel in our framework (A3). Comparison reveals that the RAK setting outperforms the Meta-Kernel setting by an improvement of 2.85 average L1 AP, further validating the efficacy of RAK in our approach. Thirdly, we investigate the optimality of the Perception Window setting in RAK by varying the number of Perception Windows to 4 and 8, respectively (A4, A5). Analysis indicates that reducing the number of Perception Windows leads to a subtle decrease in detection AP, while increasing the number of Perception Windows brings no further performance gain. Lastly, to evaluate the efficacy of VRM, we disable VRM and RP during training and inference processes (A6). Notably, decreases in detection AP are observed for all classes, particularly for vehicles. We attribute this observation to the fact that vehicles, being relatively large objects, are more susceptible to Vision Corruption.

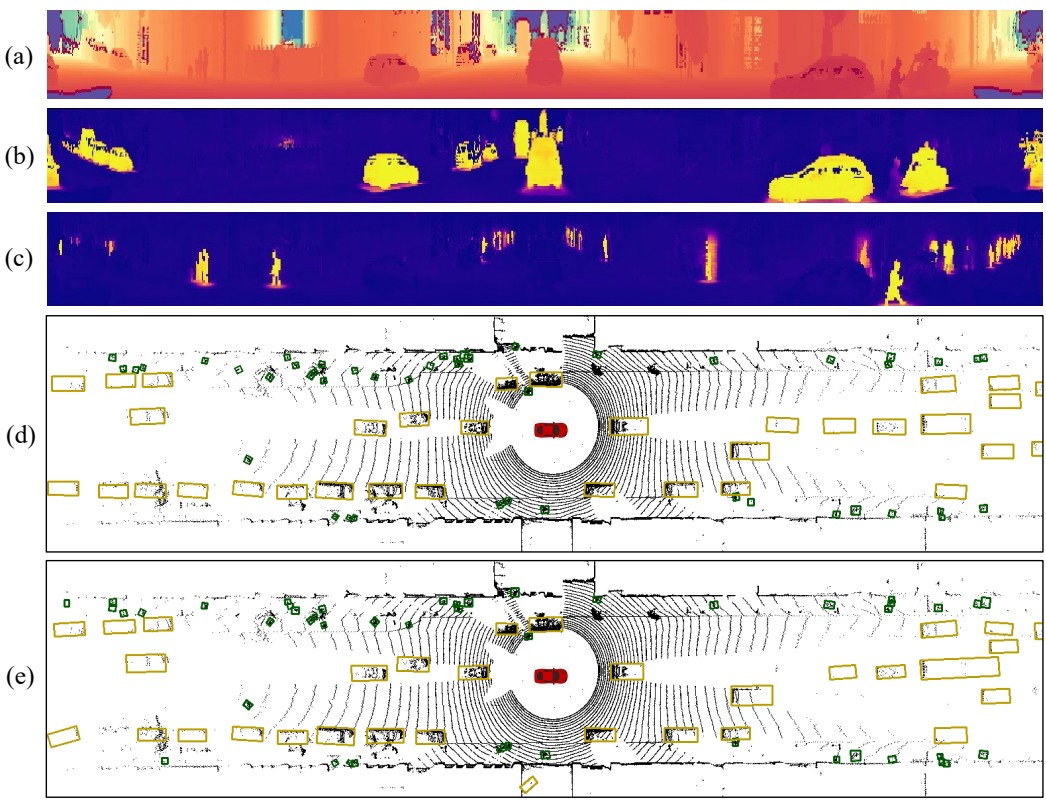

Figure 6: Qualitative detection results of RangePerception, on a validation frame of WOD. (a) Input range image. (b,c) Dense class predictions for vehicles and pedestrians. (d) Predicted boxes from BEV. (e) Ground-truth boxes from BEV. For (a-c), ego vehicle heads towards the middle of the range image. For (d,e), ego vehicle is visually highlighted in red.

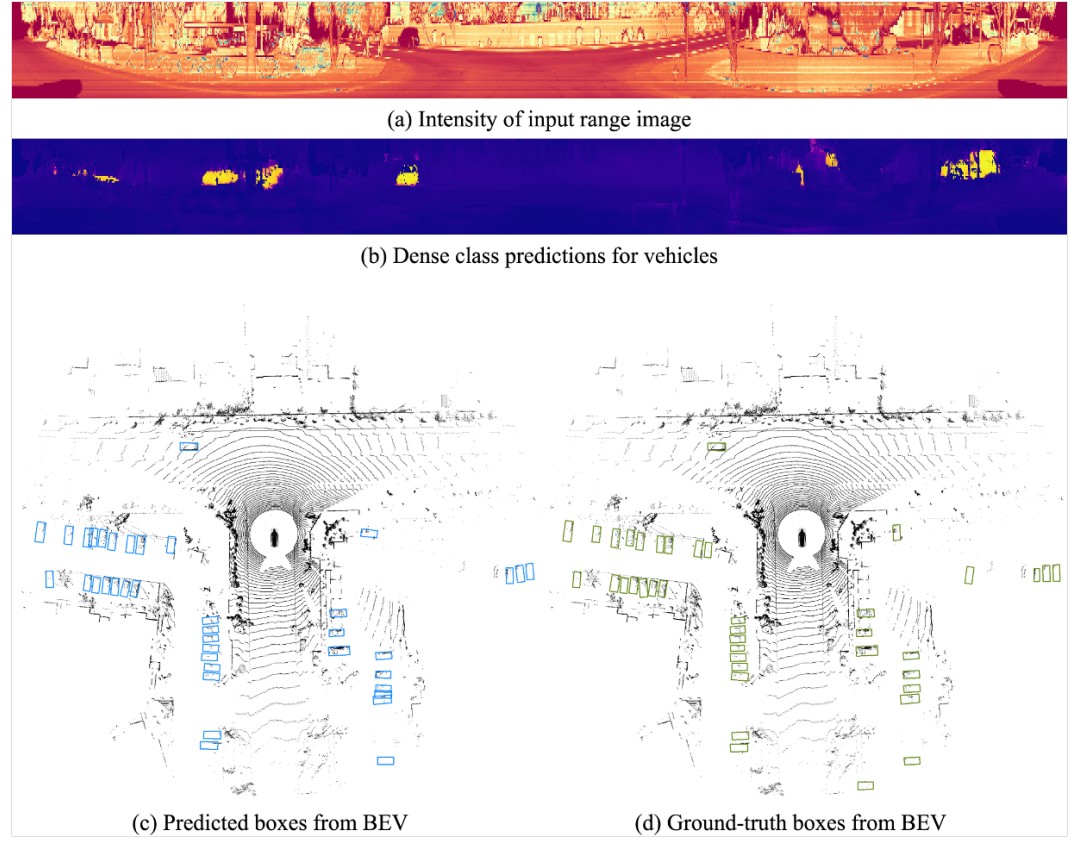

(a) Intensity of input range image

(b) Dense class predictions for vehicles

(c) Predicted boxes from BEV          (d) Ground-truth boxes from BEV

Figure 7: Qualitative vehicle detection results from RangePerception, on a validation frame of WOD. Notably, the vehicles in this frame are significantly occluded by trees and barriers. Despite these challenges, RangePerception exhibits remarkable detection performance by effectively extracting features from the range view.

**Qualitative Results.** Detection results on a WOD validation frame are visualized in Fig. 6. Notably, the sparse pedestrian foreground points are accurately preserved in RV, which eases the detection process of RangePerception. Additionally, Fig. 7 showcases RangePerception's capability of handling highly occluded foreground objects.

# 6    Limitations

RangePerception is designed to be highly compatible with point clouds generated from a single viewpoint. Similar to other existing RV-based detectors, RangePerception may not be suitable for perception tasks involving point clouds generated from multiple viewpoints, such as those obtained from multiple autonomous vehicles. However, it is important to emphasize that the occurrence of such multi-viewpoint point clouds is rare in autonomous driving scenarios. Therefore, this limitation does not affect the real-world deployment and application of RangePerception framework.

# 7    Conclusion

This paper presents RangePerception, an RV-based 3D detection framework that effectively addresses Spatial Misalignment and Vision Corruption challenges. By introducing RAK and VRM, RangePerception achieves superior detection performance on WOD, showcasing its potential for efficient and accurate real-world deployment.

## Acknowledgments and Disclosure of Funding

The research was supported by Shanghai Artificial Intelligence Laboratory, the National Key R&D Program of China (Grant No. 2022ZD0160104) and the Science and Technology Commission of Shanghai Municipality (Grant No. 22DZ1100102).

The authors express their gratitude to Zhongxuan Hu from Huazhong University of Science and Technology for the valuable assistance in refining mathematical representations in this paper.

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
