# OpenReview forum: "RangePerception: Taming LiDAR Range View for Efficient and Accurate 3D Object Detection"
_NeurIPS.cc/2023/Conference — NeurIPS 2023 poster_

### Official Review · Reviewer_GJQ1 · 2023-06-20

**Soundness:** 3 good
**Presentation:** 2 fair
**Contribution:** 3 good
**Rating:** 6
**Confidence:** 5

**Summary:**

This paper studies range-view-based LiDAR 3D object detection. It proposes Range Aware Kernel and Vision Restoration Module to handle the problems of range view representation, such as inconsistent range distribution and boundary splitting. The main contributions are as follows:

- The proposed module boosts the performance of range-view-based detection and the whole framework is very efficient, further closing the gap between range-view-based methods and BEV-based methods, which is meaningful in practical use.

- Authors are willing to release the code based on the popular OpenPCDet codebase. Given that previous works either do not open source (e.g., LaserNet, RCD) or are based on a non-mainstream framework (RangeDet), it is valuable and admirable if authors could contribute the source code.

**Strengths:**

- The proposed modules well addressed the problem in range view. In particular, Range-aware Kernel is a more elegant and efficient solution than the Range-conditioned Pyramid in RangeDet.

- Although not very extensive, the effectiveness of every proposed module is well grounded in the experiment section.

- Performance is good and reaches SoTA performance among all range-view-based methods. Compared with other BEV-based methods, its performance is also promising, especially for a single-stage detector.

- Authors are willing to release the code based on the popular OpenPCDet codebase. Given that previous works either do not open source (e.g., LaserNet, RCD) or are based on a non-mainstream framework (RangeDet), it would be valuable and admirable if authors could contribute the source code.

**Weaknesses:**

- Technical novelty is a little bit limited. The issue of spatial misalignment is in fact mentioned in RangeDet, where Meta Kernel is proposed to mitigate the problem. And the Vision Restoration is also a straightforward trick, and a similar trick (Ring CNN) is adopted in PolarNet. However, the proposed solution is indeed useful and practical. In my opinion, this point is not a fatal weakness.

- Experiments seem not extensive. This paper only contains two tables. Although these tables are informative and well validate the proposed module, these experiments are not sufficient enough to reach the bar of a high-standard conference. For example, Table 2 could be split into several small tables for more detailed ablation on each component. RAK is the core design, and it deserves a more detailed ablation on various hyper-parameters (e.g., different range partition intervals, different placed layers/positions ) or be compared with other potential alternative designs, which is necessary to reveal the inner workings.

**Questions:**

My overall impression is positive and I believe this paper has significant practical meaning in LiDAR-based 3D object detection, especially industrial use. However, I think the experiments are not sufficient enough (but well supporting the claims) to meet the bar of the high-standard NeurIPS conference. I strongly encourage authors to add more ablation experiments to offer more insight into the proposed modules. If so, I will increase my rating.

**Limitations:**

Limitations have been adequately addressed.

---

> ### Author Rebuttal · Authors · 2023-08-04
>
> Thank you for your positive recognition of our work. We hope the discussion below could address your concerns.
>
> ## Response 1.1 Extensive Ablation Study for Range Aware Kernel (Question 1 & Weakness 2)
>
> We have conducted thorough comparison experiments during the development process of the Range Aware Kernel (RAK). Below, we present the quantitative results of these experiments, which will serve as an extensive ablation study in the next version of this paper.
>
> The results of the ablation study are presented in Table A (in global response 5.1), where we assess the effectiveness of RAK.
>
> **Comparison with Other Operators** Firstly, this study investigates the impact of RAK by removing it from our framework (E1) and replacing it with a 1$\times$1 Convolution layer of equal dimension (E2). The results show a significant decrease of more than 5 average L1 AP for both cases, highlighting the crucial role of RAK in achieving strong detection performance. Secondly, we compare the performance of RAK and Meta-Kernel by replacing RAK with Meta-Kernel in our framework (E3). Comparison reveals that the RAK setting outperforms the Meta-Kernel setting by an improvement of 2.85 average L1 AP, further validating the efficacy of RAK in our approach.
>
>
> **Adjusting Number of Windows.** We investigate the optimality of the Perception Window setting in RAK by varying the number of Perception Windows (E4 - E7), where the corresponding hyper-parameters are presented in Table B. Reducing the number of Perception Windows to 2 and 4 leads to decreases in detection AP (E4, E5), indicating that an insufficient number of Perception Windows hinders the remedy for Spatial Misalignment. Increasing the number of Perception Windows to 8 and 10 brings no further performance gain (E6, E7), which we hypothesize is because an increased number of Perception Windows makes subspaces $K$ sparser and generates redundant convolution parameters.
>
> **Adjusting Placement of Windows.** We explore the optimality of the Perception Window setting in RAK by experimenting with different placements of Perception Windows (E8, E9). As is mentioned in our main paper and Table B, RAK's Perception Windows $W$ are designed as a sequence of overlapped range intervals $w_i = [r_{i1}, r_{i2}]$, where the length of each range interval $l_i = r_{i2} - r_{i1}$ gradually increases. The overlapped alignment of Perception Windows better handles the cases where objects are separated by margins of Perception Windows. For example, if a vehicle lies on the margin of $w_1 = [0, 15]$ and $w_3 = [15, 30]$, its feature can still be preserved in $w_2 = [10, 20]$. Further, the design of gradually increasing window length $l_i$ considers the fact that as the LiDAR beams reach farther, the distribution of LiDAR points becomes sparser, and our design balances the point density in each Perception Window, thus enabling more fluent feature extraction. To verify the intuitions above, experiments are conducted for non-overlapped windows (E8) and uniformly distributed windows (E9), subtle performance drops can be observed for both cases, justifying the effectiveness of our design.
>
> **Integrating RAK into Range Backbone.** Placing RAK in the mid of Range Backbone is also examined, where experiments are conducted for the case in which RAK is placed after the first convolution block of Range Backbone (E10). Qualitative result indicates that such network design leads to a minor performance degrading, which we attribute to the fact that Spatial Misalignment can be more effectively resolved at the earliest stage of feature extractor.
>
> ### Table B: Hyper-parameters of Range Aware Kernel's Ablation Study.
>
> |  | Setting | Perception Windows $W$ |
> | :---: | :---: | :---: |
> | E4 | 2 Perception Windows | $\{[0,40],[30, \infty)\}$ |
> | E5 | 4 Perception Windows | $\{[0,30],[20,50],[30,60],[50, \infty)\}$ |
> | E6 | 8 Perception Windows | $\{[0,10],[5,15],[10,20],[15,25],[25,35],[30,45],[35,60],[45, \infty)\}$ |
> | E7 | 10 Perception Windows | $\{[0,10],[5,15],[10,20],[15,25],[25,35],[30,40],[35,45],[40,50],[45,55],[50, \infty)\}$ |
> | E8 | Non-overlapped Windows | $\{[0,10],[10,20],[20,30],[30,45],[45,60],[60, \infty)\}$ |
> | E9 | Uniformly Distributed Windows | $\{[0,20],[15,35],[30,50],[45,65],[60,80],[75, \infty)\}$ |
> |  | RangePerception | $\{[0,15],[10,20],[15,30],[20,40],[30,60],[45, \infty)\}$ |
>
>
> ## Response 1.2 Comparison with Existing Works (Weakness 1)
>
> **RangeDet.** We are aware that Spatial Misalignment issue is in also mentioned in RangeDet, where Meta-Kernel is proposed to mitigate the problem. We hope to bring to your attention that we are the first work to explicitly visualize and explain Spatial Misalignment issue (main paper L72-76 \& Fig.1(e)). We also implement Meta-Kernel and compare it with RAK in our ablation study A3 (main paper L289-292). The empirical comparison demonstrates a significant 2.85 average L1 AP improvement of the RAK setting over the Meta-Kernel setting, further affirming the effectiveness of RAK in our approach.
>
> **PolarNet.** We appreciate your acknowledgment of the Ring CNN design proposed by PolarNet. We confirm that Ring CNN offers a viable solution to the Vision Corruption issue mentioned in our paper. Furthermore, we agree that our Vision Restoration Module, as proposed in our paper, is also an effective and practical approach to address the Vision Corruption issue. In the upcoming version of our paper, we will incorporate PolarNet into the related work section and provide a concise comparison between our approach and the Ring CNN design.

---

> > ### Comment · Reviewer_GJQ1 · 2023-08-10
> >
> > Thanks for the additional experiments, which absolutely make a better paper and resolve most of my concerns. Before my final rating, I have a few more questions.
> > - The performance of E3 is quite similar to RangeDet but not completely consistent. Do you implement E3 yourself or cite from anywhere else?
> > - From E4 to E5, the performance has huge leap. I'd like to listen to your analysis or opinions.
> > - I notice you are willing to share the code after acceptance. Are you planning to release other baseline methods you implemented (e.g., Meta-Kernel, ) under the same codebase? If you do, it would be a good contribution to this field.

---

> > > ### Author Response · Authors · 2023-08-11
> > >
> > > We extend our heartfelt appreciation for your insightful questions and valuable suggestions, as they will significantly enrich the quality of our manuscript. We believe that the upcoming discussion will serve to address your questions more comprehensively.
> > >
> > > ## 1. Experiment E3
> > >
> > > Extensive ablation experiment E3 is implemented, trained, and evaluated within our codebase. To maintain consistency and eliminate potential confounding factors such as differences in deep learning frameworks and hyper-parameters, we do not report RangeDet paper's performance on experiment E3. Actually, all ablation experiments in Table A are implemented, trained, and evaluated within our codebase, ensuring a comprehensive and rigorous comparative analysis.
> > >
> > > ## 2. Experiments E4 \& E5
> > >
> > > We analyze the significant performance gap between experiments E4 and E5 in the following two aspects.
> > >
> > > **Remedy for Spatial Misalignment.** Our analysis indicates that the limited number of Perception Windows adopted in E4 is insufficient to effectively address Spatial Misalignment. Referring to the hyper-parameter settings detailed in Table B, it becomes evident that E4 is unable to rectify any Spatial Misalignment issues that lie within the range interval of $[30, \infty)$. Conversely, E5, by further subdividing this interval into two distinct subspaces, namely $\{[30, 60], [50, \infty)\}$, exhibits the capability to handle a substantial portion of Spatial Misalignments occurring within the $[30, \infty)$ range interval. In essence, the finer placement of Perception Windows in E5 empowers it to effectively resolve a larger portion of Spatial Misalignment issues within the range image. This finer granularity contributes significantly to the superior performance of E5 compared to E4.
> > >
> > > **Number of Convolution Kernels.** As depicted in our main paper's Figure 2, the transformed subspaces $K' \in \mathbb{R} ^ {{m} \times {n} \times {8l}}$ undergo non-linear feature extraction through the Range Backbone, where $l$ represents the number of Perception Windows in RAK. We denote the initial 2D Convolution Layer in the Range Backbone as $\mathcal{C}$, which produces an output feature map with a dimension of $d$, denoted as $H \in \mathbb{R} ^ {{m} \times {n} \times {d}}$. Notably, the 2D Convolution Layer $\mathcal{C}$ encompasses a total of $8l \times d$ fully connected kernels. Our ablation experiments, as detailed in Table A, utilize $l = 2$ for E4 and $l = 4$ for E5. Consequently, it is evident that the 2D Convolution Layer $\mathcal{C}$ in E5 employs double the number of kernels compared to E4, thereby yielding enhanced representation power in feature extraction. This increase in learnable kernels contributes to the notably stronger performance observed in E5.
> > >
> > > To summarize, the divergence in performance observed between experiments E4 and E5 can be attributed to E5's superior handling of Spatial Misalignment and its increased representation power through a larger count of convolution kernels.
> > >
> > > ## 3. Open-sourced Codebase
> > >
> > > We confirm that the codebase of RangePerception will be open-sourced after the acceptance of this paper. Our open-sourced codebase will also include our PyTorch implementation of RangeDet, encompassing both Meta-Kernel Convolution and Range Conditioned Pyramid. Meanwhile, our PyTorch implementation of single-frame FCOS-LiDAR will be made public.

---

> > > > ### Comment · Reviewer_GJQ1 · 2023-08-12
> > > > **Technical solid paper, worthy of acceptacne**
> > > >
> > > > After reading the responses to all reviewers and engaging in further discussion, the authors have addressed most of my concerns. I believe it is a technically solid paper and a good improvement on the previous range-based detectors. So I increase my rating to 6.
> > > >
> > > > Authors are supposed to add the following stuff into the final version:
> > > > 1. Table A (maybe a brief version)
> > > > 2. Link for code releasing
> > > > 3. Proper credits to previous methods, for example, Ring RCNN in PolarNet and some similar components in RangeDet. The authors should make the differences between them and the components in this paper more clear.

---

> > > > > ### Author Response · Authors · 2023-08-13
> > > > >
> > > > > Thank you for your thoughtful review and for recognizing our work. We're glad you find our paper a meaningful advancement in the realm of range-based detectors.
> > > > >
> > > > > We will certainly include the following in the final version:
> > > > >
> > > > > 1. Our extensive ablation study, including Table A and corresponding discussion.
> > > > >
> > > > > 2. A link to the open-sourced codebase of RangePerception.
> > > > >
> > > > > 3. Appropriate credits for previous methods, including similar components proposed in PolarNet and RangeDet, and highlight the distinct features of our approach compared to these methods in the final version. This clarification will provide a more transparent understanding of our contributions within the context of existing work.
> > > > >
> > > > > Thank you for your continued support in refining our work.

---

### Official Review · Reviewer_PXtx · 2023-07-03

**Soundness:** 3 good
**Presentation:** 3 good
**Contribution:** 3 good
**Rating:** 6
**Confidence:** 4

**Summary:**

The paper presents an efficient and accurate RV-based 3D object detection framework called *RangePerception*. The framework addresses two critical challenges impeding the performance of existing RV-based methods: 1) a natural domain gap between the 3D world coordinate used in output and 2D range image coordinate used in input, generating difficulty in information extraction from range images; and 2) native range images suffer from vision corruption issue, affecting the detection accuracy of objects located on the margins of the range images. To address these challenges, two novel algorithms named Range Aware Kernel (RAK) and Vision Restoration Module (VRM) are proposed. With the help of RAK and VRM, RangePerception achieves higher averaged L1/L2 AP compared to previous state-of-the-art RV-based method RangeDet on Waymo Open Dataset. For the first time as an RV-based 3D detection method, RangePerception achieves slightly superior averaged AP compared with the well-known BEV-based method CenterPoint and has an inference speed that is 1.3 times as fast as CenterPoint. The paper’s contributions include the RangePerception Framework, Range Aware Kernel and Vision Restoration Module.

**Strengths:**

**Originality**: The paper introduces two modules, Range Aware Kernel (RAK) and Vision Restoration Module (VRM), to address the challenges of existing RV-based methods. The proposed RangePerception framework is original in its approach to efficiently and accurately detect 3D objects using LiDAR range view.

**Quality**: The paper presents a thorough analysis of the challenges impeding the performance of existing RV-based methods and proposes solutions to address these challenges. The paper provides ablation studies to demonstrate the effectiveness of each component. Supplementary Material is well written.


**Weaknesses:**

**Major Issues**:

1. In table 1, it seems that many related works [1-2] are missing. They also focus on Efficient and Accurate 3D Object Detection, and the authors should also compare with them.
2. Experiments are insufficient. It's interesting to see the performance on other datasets (e.g., KITTI and nuScenes). For nuScenes dataset, they use a 32-beam LiDAR (KITTI and WOD both are 64-beam). Does the proposed RV-based method still performs well for extremely low resolution range images (32 or even 16-beam LiDAR)?

**Minor Issues**:
1. Inconsistent definition and dimension of *range image* in Algorithm 1 L.1 (range image $I \in R^{m \times n \times 8}$) and in Paper L.119-L.121 (m x n matrix).
2. Please add citations in Tab.1 first column (*Method* column).
3. Although the illustration on Redundancy Pruning in Fig. 5(c) is straight forward, it's better to also include the real range image visualization for Redundancy Pruner in Fig. 5 (a) or (b).
4. It could be easier to follow introducing the **Related Work** section ahead of the **Methodology**.


[1] Liu, Zhijian, Haotian Tang, Alexander Amini, Xinyu Yang, Huizi Mao, Daniela Rus, and Song Han. "BEVFusion: Multi-Task Multi-Sensor Fusion with Unified Bird's-Eye View Representation." In ICRA. 2023.

[2] Sun, Pei, Weiyue Wang, Yuning Chai, Gamaleldin Elsayed, Alex Bewley, Xiao Zhang, Cristian Sminchisescu, and Dragomir Anguelov. "Rsn: Range sparse net for efficient, accurate lidar 3d object detection." In CVPR. 2021.


**Questions:**

Range view detection could be inevitably affected by **occlusion** and **scale variation** due to the spherical projection. Does the proposed method consider this?

**Limitations:**

The limitations are adequately included in the Supplementary Material.

---

> ### Author Rebuttal · Authors · 2023-08-07
>
> We appreciate your recognition of our work. We hope the following discussion adequately addresses your concerns.
>
> ## Response 2.1 Occlusion and Scale Variation Phenomena (Question 1)
>
> As is illustrated in our main paper L202-204 \& Fig.2,  RangePerception learns the pixel-wise dense predictions, namely classification $C$, regression $B$, and confidence $U$. It is straightforward to observe from our main paper Fig.2 \& Fig.6(b, c), where the class prediction head of RangePerception accomplishes the same task as foreground segmentation. Such segmentation-based learning principle adequately addresses the issue of occlusion and scale variation: even though some objects have limited foreground points due to far distance or strong occlusion, their geometric attributes can still be learned and predicted via the segmentation-based paradigm.
>
>
> ## Response 2.2 Performance on nuScenes Dataset  (Major Weakness 2)
>
> The results on nuScenes dataset are presented in this section.
>
> **Baseline Methods.** As shown in Table C, state-of-the-art BEV-based, MV-based, and RV-based detectors are selected as baseline methods. All BEV-based and MV-based baselines are trained and evaluated with OpenPCDet's official PyTorch implementation. We reimplement and examine RangeDet as described in our main paper L269-270. Since FCOS-LiDAR reports results on nuScenes without releasing the code, we reimplement FCOS-LiDAR in our codebase, and achieve similar single-frame results as reported in their paper.
>
> **Experiment Settings.** Training and evaluation are conducted with single-frame setting on nuScenes dataset. All models are trained with $20$ epochs on nuScenes training set, where batch size is $32$ and frame sampling rate is $100\%$. Inference speed is examined with one NVIDIA 3090Ti GPU with batch size set to $1$.
>
> **Main Results.** The quantitative performance of RangePerception is surprisingly excellent, surpassing all baseline methods in terms of NDS and average AP. Particularly, our method exhibits outstanding performance in detecting small objects, such as pedestrians, motors, bicycles, and traffic cones. This observation reaffirms the analysis presented in our main paper L277-280 \& L300-302, where the range-view representation better preserves visual features of small objects, while voxelization introduces quantization errors to the originally sparse foreground points. Additionally, RangePerception demonstrates superior performance in detecting construction vehicles, which we attribute to our segmentation-based learning paradigm being more sensitive to the distinctive shape of construction vehicles. The remarkable overall performance and balanced perception ability across all object categories further validate that the RangePerception framework effectively handles scale variation.
>
> ### Table C: Detection performance measured by NDS and AP on nuScenes validation set. All experiments are conducted under single-frame setting. C.V. stands for construction vehicle, and T.C. stands for traffic cones.
>
> | Method             | View | Stage | NDS   | AP    | Car   | Truck | Bus   | Trailer | C.V. | Ped   | Motor | Bicycle | T.C.  | Barrier | FPS   |
> |--------------------|------|-------|-------|-------|-------|-------|-------|---------|------|-------|-------|---------|-------|---------|-------|
> | Second             | B    | one   | 50.48 | 50.03 | 75.24 | 46.04 | 54.17 | 47.99   | 16.73| 69.50 | 42.07 | 19.06   | 67.33 | 62.12   | 16.72 |
> | PointPillar        | B    | one   | 50.02 | 49.27 | 74.70 | 45.46 | 53.23 | 45.99   | 16.46| 68.71 | 40.96 | 18.73   | 66.51 | 61.96   | 23.89 |
> | CenterPoint        | B    | one   | 54.95 | 53.92 | 78.19 | 47.53 | 55.92 | 49.54   | 16.96| 77.09 | 49.99 | 22.62   | 70.98 | **65.69**   | 19.75 |
> | PV-RCNN            | B+P  | two   | 54.23 | 53.12 | 80.75 | 49.09 | 57.65 | 51.26   | 17.51| 74.84 | 47.27 | 21.54   | 66.78 | 64.47   | 1.85  |
> | Part-$A^2$-anchor  | B+P  | two   | 54.67 | 54.10 | **81.10** | **49.30** | **57.91** | **51.45**   | 21.58| 76.16 | 48.13 | 21.82   | 67.96 | 65.61   | 4.69  |
> | FCOS-LiDAR         | R    | one   | 53.41 | 53.43 | 72.95 | 42.33 | 46.95 | 43.31   | 25.56| 74.99 | 60.35 | 34.61   | 70.29 | 62.74   | 25.23 |
> | RangeDet           | R    | one   | 54.39 | 54.17 | 73.41 | 42.83 | 48.15 | 44.33   | 26.37| 75.46 | 60.58 | 34.58   | 71.52 | 64.50   | 19.87 |
> | RangePerception    | R    | one   | **55.65** | **55.31** | 75.68 | 43.91 | 48.71 | 44.93   | **26.51**| **78.80** | **60.61** | **35.90** | **72.93** | 65.10   | **26.17** |
>
> ## Response 2.3 Comparison with BEVFusion and RSN (Major Weakness 1)
>
> We will cite and compare with these two methods in the next version of our paper. Meanwhile, we hope to point out that the problem settings of BEVFusion and RangePerception are naturally different: (1) BEVFusion is a multi-modal joint-LiDAR-camera detection method; (2) RangePerception, on the other hand, is a single-modal detection method based on LiDAR signal.
>
> ## Response 2.4 Range Image Definition (Minor Weakness 1)
>
> The definition of range image $I \in \mathbb{R} ^ {{m} \times {n} \times {8}}$ and range matrix $R \in \mathbb{R} ^ {{m} \times {n}}$ will be specifically clarified in the next version of this paper.
>
> ## Response 2.5 Visualization for Redundancy Pruner (Minor Weakness 3)
>
> We would like to clarify that Redundancy Pruner operates on feature space $F^r$ (described in main paper L191) instead of range image $I$, which is the reason why we choose to visualize redundancy pruner with pseudo feature in main paper Fig.5(c). That being said, we can include a new figure which visualizes how Redundancy Pruner operates on the real feature space, if you believe that would be helpful.
>
> ## Response 2.6 Placement of Related Work (Minor Weakness 2 & 4)
>
> Citations will be added to Table 1 in the next version of this paper. Meanwhile, Related Work section will be placed ahead of the Methodology section.

---

> > ### Comment · Reviewer_PXtx · 2023-08-14
> > **Decision to Elevate My Rating**
> >
> > Thank you for the additional experiments and rebuttals.
> >
> > The rebuttal has addressed most of my questions and concerns, especially regarding performance on low-resolution datasets (nuScenes) and Occlusion & Scale Variation Phenomena.
> >
> > Therefore, I am inclined to accept this paper and raise my rating.

---

> > > ### Author Response · Authors · 2023-08-15
> > >
> > > We deeply value your meticulous review and are pleased that our responses have effectively addressed your questions and concerns. Your willingness to raise your rating and accept the paper is truly appreciated. We extend our profound appreciation for your insightful questions and invaluable suggestions, as they undoubtedly contribute to the elevation of our manuscript's scholarly caliber.

---

### Official Review · Reviewer_2sVP · 2023-07-04

**Soundness:** 3 good
**Presentation:** 3 good
**Contribution:** 3 good
**Rating:** 6
**Confidence:** 5

**Summary:**

This paper proposes RangePerception, an RV-based 3D detection framework that aims to address the issues of Spatial Misalignment and Vision Corruption in range-view representation. The Range Aware Kernel (RAK) disentangles the range image space into multiple sub-spaces, and overcomes the Spatial Misalignment issue by enabling independent feature extraction from each subspace. The Vision Restoration Module (VRM) builds an extended spherical space by pre-defining a restoration angle to restore visual features originally corrupted by the LiDAR’s sampling process on both sides of range image.

**Strengths:**

1. This paper presents a novel high-performing RV-based 3D object detection framework.
2. The specially designed RAK and VRM modules effectively solve the Spatial Misalignment and Vision Corruption problems in range-view representation.
3. The paper is well-organized and clearly written especially in the introduction part.

**Weaknesses:**

1. The paper only provides the results on the Waymo validation set, lacking the results on the Waymo test set.
2. Lack of evaluation based on different distances like RangeDet.
3. No experiments are conducted on other mainstream datasets such as nuScenes and KITTI.
4. Lack of ablation studies on restoration angle hyper-parameters.

**Questions:**

1. For RV-based 3D detection, the scale-variation and occlusion issues are also two major challenges. Is the proposed method helpful to them?
2. For Range Aware Kernel, window partitioning inevitably leads to the problem of separating one object into different sub-spaces. Can you carefully analyze this issue?
3. After disentangling into multiple sub-spaces, the range image will become extremely sparse. Why does the feature extraction network still use 2D dense convolution instead of 2D sparse convolution?

**Limitations:**

The authors discuss the limitations and there are no potential negative societal impacts.

---

> ### Author Rebuttal · Authors · 2023-08-07
>
> Thank you for recognizing our work. We aim to address your concerns adequately in the following discussion.
>
> ## Response 3.1 Occlusion and Scale Variation Phenomena (Question 1)
>
> This topic is carefully discussed in Response 2.1 to reviewer PXtx.
>
> ## Response 3.2 Performance on nuScenes Dataset (Weakness 3)
>
> We include performance on nuScenes dataset in Response 2.2 to reviewer PXtx.
>
> ## Response 3.3 Separation of Objects (Question 2)
>
> As is introduced in our main paper L259, RAK's Perception Windows $W$ are designed as a sequence of overlapped range intervals $w_i = [r_{i1}, r_{i2}]$. The overlapped alignment of Perception Windows better handles the cases where objects are separated by margins of Perception Windows. For example, if a vehicle lies on the margin of $w_1 = [0, 15]$ and $w_3 = [15, 30]$, its feature can still be preserved in $w_2 = [10, 20]$. The benefit brought by overlapped window alignment is further validated by our extensive ablation study E8, presented in Response 1.1 to Reviewer GJQ1.
>
> ## Response 3.4 Dense 2D Convolution (Question 3)
>
> As illustrated in our main paper Fig. 2, transformed subspaces $K' \in \mathbb{R} ^ {{m} \times {n} \times {8l}}$ are processed by Range Backbone for non-linear feature extraction. We denote the first 2D Convolution Layer in Range Backbone as $\mathcal{C}$, whose output dimension is $d$. Subsequently, the output feature from $\mathcal{C}$ can be represented as $H \in \mathbb{R} ^ {{m} \times {n} \times {d}}$. Note that 2D Convolution Layer $\mathcal{C}$ consists of $8l \times d$ number of fully connected kernels. Due to the fully connected nature of $\mathcal{C}$, feature $H = \mathcal{C}(K')$ is a dense tensor, despite the sparsity of $K'$. As a result, the input and output features of each consecutive layer in Range Backbone are also dense tensors. It is therefore unfeasible to build Range Backbone with sparse 2D Convolutions.
>
> ## Response 3.5 Choice of Restoration Angle (Weakness 4)
>
> We randomly sample 200 frames of range images from Waymo Open Dataset. By visualization, we find that restoration angle $\delta = 0.086\pi$ is sufficient to resolve the Vision Corruption issues that occur in the sampled frames, without introducing unnecessary redundancy.
>
> An extensive ablation study for Vision Restoration Module is also conducted. As presented in Table D, ablations are experimented by disabling VRM and RP (E11), reducing restoration angle to $0.043\pi$ (E12), and increasing restoration angle to $0.172\pi$ (E13). Comparison demonstrates that all three sizes of restoration angle $\delta$ boost the detection accuracy, while the selected hyper-parameter $\delta = 0.086\pi$ generates the most ideal numerical results.
>
> ### Table D: Ablation study of Vision Restoration Module, measured by 3D AP/APH on WOD validation set.
>
> |        | Setting               | Vehicle     | Vehicle | Pedestrian   | Pedestrian  | Cyclist      | Cyclist      | Average      | Average      |
> |:--------:|:-----------------------:|:-------------:|:---------:|:--------------:|:---------:|:--------------:|:---------:|:--------------:|:---------:|
> |        |                       | L1          | L2      | L1           | L2      | L1           | L2      | L1           | L2      |
> | E11    | without VRM & RP      | 72.50/71.97 | 66.40/65.91 | 80.20/76.08 | 72.27/68.53 | 70.31/68.92 | 68.74/67.42 | 74.34/72.32 | 69.13/67.29 |
> | E12    | $\delta = 0.043\pi$            | 73.46/73.03 | 66.45/65.93 | 80.22/76.10 | 72.28/68.53 | **70.33/68.93** | 68.74/67.43 | 74.67/72.69 | 69.15/67.30 |
> | E13    | $\delta = 0.172\pi$            | 73.60/73.09 | 66.46/65.99 | **80.24/76.12** | 72.29/68.53 | 70.32/68.92 | **68.75/67.43** | 74.72/72.71 | 69.16/67.31 |
> |        | RangePerception       | **73.62/73.11** | **66.47/66.00** | **80.24/76.12** | **72.29/68.54** | **70.33/68.93** | **68.75/67.43** | **74.73/72.72** | **69.17/67.32** |
>
> ## Response 3.6 Evaluation based on Different Distances  (Weakness 2)
>
> Shown in Table E, we conduct an extensive evaluation on the WOD validation set, measuring the detection performance in terms of L1 3D AP for different distance ranges. For comparison, we use RangeDet as the baseline method. RangePerception demonstrates superior performance compared to RangeDet across different distances for all classes. The results validate the effectiveness of RangePerception in handling diverse scenarios and addressing scale variation issues.
>
> ### Table E: Detection performance measured by L1 3D AP on WOD validation set, based on different distances.
>
> | Class       | Method           | Overall | 0 - 30m | 30 - 50m | 50m - $\infty$  |
> |-------------|------------------|:---------:|:---------:|:---------:|:---------:|
> | Vehicle     | RangeDet         | 72.85   | 87.96   | 69.03   | 48.88   |
> | Vehicle     | RangePerception  | **73.62**   | **88.79**   | **69.77**   | **49.45**   |
> | Pedestrian  | RangeDet         | 75.94   | 82.20   | 75.39   | 65.74   |
> | Pedestrian  | RangePerception  | **80.24**   | **86.87**   | **79.64**   | **69.36**   |
> | Cyclist     | RangeDet         | 65.67   | 79.33   | 55.80   | 45.00   |
> | Cyclist     | RangePerception  | **70.33**   | **82.95**   | **59.35**   | **50.19**   |
>
> ## Response 3.7 Performance on Waymo Test Set (Weakness 1)
>
> Quantitative and qualitative results on Waymo test set will be presented and discussed in the next version of this paper.

---

> > ### Comment · Reviewer_2sVP · 2023-08-20
> > **Good feedback**
> >
> > The authors well address my concerns in the feedback and I thus decide to upgrade my score to Weak Accept.

---

> > > ### Author Response · Authors · 2023-08-21
> > >
> > > We are grateful to you for recognizing our efforts in addressing your concerns during the response process. Your feedback has been instrumental in enhancing the quality of our work, and we look forward to continuing to meet your expectations in the final version of our paper.

---

### Official Review · Reviewer_ZtTP · 2023-07-08

**Soundness:** 3 good
**Presentation:** 3 good
**Contribution:** 2 fair
**Rating:** 6
**Confidence:** 5

**Summary:**

This paper proposed a new range view based 3D object detection algorithm, which achieved on-par performance compared to sota BEV/voxel based algorithm.

**Strengths:**

1. The paper starts from first principle, analyzes the possible information losses during the lidar data collection/analysis/transformation process, and comes up with corresponding strategies to enhance/improve range view perception performance.
2. The paper comes up with effective strategies for training range view detection models, and achieved on-par performance with sota voxel based methods, and runs much faster.

**Weaknesses:**

1. The core idea of RAK is to divide the computation window based on the range. This is hardly entirely new/novel for range based methods, range conditioned convolution/detection has been widely studied before. The reviewer implemented very similar strategies but have not found experimental improvements in the range view detection task.
2. The VRM module does not demonstrated experimental benefits.

**Questions:**

1. Modern lidar sensors usually carry multiple return signals with each laser pulse. For example, WOD dataset has multiple returns, each return forming a unique range image. It is unclear how this framework takes multi-return into consideration.

**Limitations:**

No negative societal impact is noticed to the best of reviewer's knowledge.

---

> ### Author Rebuttal · Authors · 2023-08-08
>
> We appreciate your positive acknowledgment of our work. We hope that the subsequent discussion effectively addresses the concerns you have raised.
>
> ## Response 4.1 Incorporation of Multi-return Range Images (Question 1)
>
> The incorporation of multi-return LiDAR data is extensively investigated and evaluated during the development of RangePerception. Actually, the LiDAR data in Waymo Open Dataset (WOD) consists of only two returns, namely first return and second return. First-return and second-return range images of several sampled frames in WOD are visualized in Fig.1 of our global response PDF file. Through this set of visualization, it is evident that the first return of WOD LiDAR is significantly more compact and informative; in contrast, the second return is highly sparse and contains little information.
>
> To reaffirm this conclusion, we conduct a random sampling of 1000 foreground objects for each category in WOD. We then calculate the average number of first-return and second-return foreground points within these objects. The results, presented in Table F, clearly demonstrate that the second return is significantly sparser compared to the first return. Specifically, for vehicle category, the second-return points account for only 2.17\% of the total points. This observation reinforces the conclusion that the first return contains more informative data, while the second return is sparse and provides limited information.
>
> ### Table F: Statistics of sampled objects in WOD, in terms of first-return and second-return foreground points. The average numbers of first-return and second-return foreground points in each object are shown in the first and second data column. Additionally, the ratio of second-return points over all foreground points is shown in the third data column.
>
> | Class       | Avg. 1st Points | Avg. 2nd Points | 2nd Points Ratio |
> |:-----------:|:----------------:|:-----------------:|:------------------:|
> | Vehicle     | 613.97          | 13.68           | 2.17\%            |
> | Pedestrian  | 95.46           | 5.60            | 5.54\%            |
> | Cyclist     | 155.79          | 3.93            | 2.46\%            |
>
> Indeed, the sparsity and limited information of the second return in the LiDAR data lead previous range-view-based detectors, such as RangeDet, to exclusively utilize the first return as both input data and for target assignment. To make a fair comparison with RangeDet, RangePerception is also implemented solely based on first-return range images. It is worth pointing out that in our main paper Table 1, the results of RangeDet and RangePerception are achieved solely based on the first-return LiDAR input, while other baseline methods are evaluated based on both returns of LiDAR input. Notably, RangePerception achieves state-of-the-art performance without the help of second-return data.
>
> Despite the facts above, explorations are also made to exploit the information contained in second-return range image. We adopt a straightforward approach that concatenates the second-return range image $S \in \mathbb{R} ^ {{m} \times {n} \times {8}}$ with transformed subspaces $K' \in \mathbb{R} ^ {{m} \times {n} \times {8l}}$, generating an augmented subspace tensor $A \in \mathbb{R} ^ {{m} \times {n} \times {8(l+1)}}$. Augmented subspace tensor $A$ is further input to Range Backbone for feature extraction. As shown in Table G, the performance gain introduced by this operation is relatively marginal, where the improvement to average L1 AP is only 0.08. Since the second-return data is initially sparse and hardly suffers from Spatial Misalignment, we do not attempt to process second-return data with Range Aware Kernel.
>
> ### Table G: Comparision study of two-return range image input, measured by 3D AP/APH on WOD validation set. Experiments are conducted under single-frame setting.
>
> | Method | View | Stage | Vehicle | Vehicle | Pedestrian | Pedestrian | Cyclist | Cyclist | Average | Average | FPS |
> | :---: | :---: | :---: | :---: | :---: | :---: | :---: | :---: | :---: | :---: | :---: | :---: |
> |  |  |  | L1 | L2 | L1 | L2 | L1 | L2 | L1 | L2 |  |
> | First-return Input |  R  | one |  73.62/73.11  |  66.47/66.00  |  80.24/76.12  |  72.29/68.54  |  70.33/68.93  |  68.75/67.43  |  74.73/72.72  |  69.17/67.32  | **45.85** |
> | Two-return Input |  R  | one |  **73.71/73.20**  |  **66.55/66.08**  |  **80.27/76.15**  | **72.32/68.57**  |  **70.45/69.05**  |  **68.78/67.45**  |  **74.81/72.80**  |  **69.27/67.37**  | 44.73 |
>
> We will include the analysis and experiment results above in the supplementary material of our paper. Since the incorporation of second-return range image is not the focus of this study, we leave methodology exploration for the utilization of second return to future work.
>
> ## Response 4.2 Range Aware Kernel (Weakness 1)
>
> We address your concerns regarding Range Aware Kernel in our global response 5.2.
>
> ## Response 4.3 Vision Restoration Module (Weakness 2)
>
> The experimental advantages resulting from the incorporation of the Vision Restoration Module (VRM) are elucidated through ablation study A6, as discussed in our main paper at L296-299. In accordance with the findings presented in A6, the integration of VRM leads to an enhancement in L1 AP/APH for vehicle detection, with improvements of 1.12/1.14. Notably, the observed enhancements in the cases of pedestrian and cyclist detection are comparatively limited. We attribute this observation to the fact that pedestrians and cyclists, being relatively small objects, are less susceptible to Vision Corruption.

---

### Author Rebuttal · Authors · 2023-08-08

Thank you for acknowledging our work positively. We are pleased to provide the supplemental responses for part of the individual responses.

## Response 5.1 Supplemental Response for Reviewer GJQ1

### Table A: Ablation study of Range Aware Kernel, measured by 3D AP/APH on WOD validation set.

|  | Setting |   Vehicle   |   Vehicle   | Pedestrian | Pedestrian | Cyclist | Cyclist | Average | Average |
| :---: | :---: | :---: | :---: | :---: | :---: | :---: | :---: | :---: | :---: |
|  |  | L1 | L2 | L1 | L2 | L1 | L2 | L1 | L2 |
| E1 | without RAK | 70.46/70.01 | 63.45/62.89 | 72.86/68.88 | 64.56/60.72 | 63.63/62.43 | 61.44/60.25 | 68.98/67.11 | 63.18/61.29 |
| E2 | 2D Convolution | 70.48/70.03 | 63.46/62.90 | 72.86/68.87 | 64.56/60.71 | 63.64/62.43 | 61.45/60.26 | 68.99/67.11 | 63.16/61.25 |
| E3 | Meta-Kernel | 72.95/72.43 | 65.98/65.37 | 75.95/71.96 | 67.63/63.90 | 66.76/65.46 | 64.57/63.38 | 71.88/69.95 | 66.06/64.21 |
| E4 | 2 Perception Windows | 71.34/70.85 | 64.31/63.75 | 73.63/69.65 | 65.37/61.48 | 65.17/64.09 | 63.10/61.95 | 70.04/68.20 | 64.26/62.39 |
| E5 | 4 Perception Windows | 73.13/72.62 | 65.97/65.51 | 80.15/76.03 | 72.17/68.42 | 70.15/68.85 | 68.42/67.10 | 74.48/72.50 | 68.85/67.01 |
| E6 | 8 Perception Windows | 73.59/73.08 | 66.45/65.98 | 80.25/76.13 | 72.30/68.56 | 70.31/68.92 | 68.74/67.42 | 74.72/72.71 | 69.16/67.32 |
| E7 | 10 Perception Windows | 73.58/73.07 | 66.43/65.97 | **80.26/76.14** | **72.32/68.57** | 70.29/68.89 | 68.72/67.41 | 74.71/72.70 | 69.16/67.32 |
| E8 | Non-overlapped Windows | 72.14/71.53 | 66.07/65.48 | 80.03/75.97 | 72.09/68.37 | 67.12/65.88 | 65.34/64.15 | 73.09/71.13 | 67.83/66.00 |
| E9 | Uniformly Distributed Windows | 73.12/72.86 | 66.12/65.52 | 79.88/75.79 | 71.91/68.13 | 69.45/68.04 | 67.82/66.51 | 74.15/72.23 | 68.62/66.72 |
| E10 | RAK After First Conv Block | 73.61/73.10 | 66.45/65.98 | 79.84/75.78 | 71.88/68.10 | 69.41/68.01 | 67.79/66.48 | 74.29/72.30 | 68.71/66.85 |
|  | RangePerception | **73.62/73.11** | **66.47/66.00** | 80.24/76.12 | 72.29/68.54 | **70.33/68.93** | **68.75/67.43** | **74.73/72.72** | **69.17/67.32** |

## Response 5.2 Supplemental Response for Reviewer ZtTP

We are aware that previous works such as Cylinder3D and PolarNet divide computation windows based on range. However, the computation windows of previous works are designed as a set of non-overlapped intervals. Unlike prior approaches, Range Aware Kernel (RAK) incorporates a distinct innovation by adopting a set of overlapped intervals as part of its window structure design. This strategic choice ensures that objects situated at the margins of Perception Windows are not subject to information loss, thereby enhancing the fidelity of feature extraction. We believe this nuanced design, illustrated in Fig.3(a) \& L258-260 of our main paper, fundamentally differentiates RAK from previous methods.

Furthermore, it's noteworthy that Cylinder3D and PolarNet are primarily developed for segmentation tasks operating on point views. RAK, on the other hand, is purposefully tailored for the range-view-based object detection task. This tailored focus underscores RAK's pioneering role in addressing Spatial Misalignment challenges within this specific context.

Moreover, as illustrated in Response 1.1 to Reviewer GJQ1, thorough window structure designing and hyper-parameter tuning are the key enabler for the optimal functionality of RAK. Both factors significantly impact RAK's performance, as demonstrated by our extensive ablation study in main paper L286-296 \& Response 1.1.

We recognize the necessity to address the concerns of novelty and experimental improvements thoroughly. We will carefully compare our method with Cylinder3D and PolarNet in the next version of our paper. Our supplementary material will include an in-depth analysis of the hyper-parameter tuning process, substantiated by comprehensive experiments that showcase the improvements attainable through RAK when the optimal configuration is achieved. The discussions above will facilitate a more nuanced understanding of RAK's potential and its applicability within the broader context of range-view-based methods.

In conclusion, we are committed to refining our manuscript to more accurately represent the significance and effectiveness of the Range Aware Kernel, and we are grateful for your guidance in this endeavor.

---

### Decision · Program_Chairs · 2023-09-21

**Decision:**

Accept (poster)

**Comment:**

This paper finally received all four positive scores. In the first round, reviewers raised some critical questions about the novelty against existing works, the results on more datasets, and some experiment analyses. The authors carefully prepared the rebuttal and provided detailed one-to-one responses. After discussion, the reviewers agree that most concerns are well addressed, and the scores converge to accept. In summary, after revision, this paper is qualified to be published on the NeurIPS. The final recommendation is Accept.